# High Prevalence of the Lung Ultrasound Interstitial Syndrome in Systemic Sclerosis Patients with Normal HRCT and Lung Function—A Pilot Study

**DOI:** 10.3390/jcm13102885

**Published:** 2024-05-14

**Authors:** Camille Mercier, Benjamin Thoreau, Thomas Flament, Sylvie Legué, Arthur Pearson, Stephanie Jobard, Sylvain Marchand-Adam, Laurent Plantier, Elisabeth Diot

**Affiliations:** 1Service de Médecine Interne et Immunologie Clinique, Centre de Compétence Maladies Systémiques Auto-Immunes Rares, CHRU Tours, 37000 Tours, France; s.jobard@chu-tours.fr (S.J.); elisabeth.diot@univ-tours.fr (E.D.); 2Service de Médecine Interne, Centre de Référence Maladies Systémiques Auto-Immunes Rares d’Ile de France, Assistance Publique-Hôpitaux de Paris (AP-HP), 75610 Paris, France; benjamin.thoreau@aphp.fr; 3Institut Cochin, Inserm U1016, CNRS UMR 8104, Université Paris Cité, 75006 Paris, France; 4Service de Pneumologie et Explorations Fonctionnelles Respiratoires, Centre de Référence Maladies Pulmonaires Rares, CHRU Tours, 37000 Tours, France; t.flament@chu-tours.fr (T.F.); s.legue@chu-tours.fr (S.L.); s.marchandadam@univ-tours.fr (S.M.-A.); laurent.plantier@univ-tours.fr (L.P.); 5Lung Ultrasound Working Group (G-ECHO), Société de Pneumologie de Langue Française, Île-de-France, 75935 Paris, France; 6Service de Radiologie, CHRU Tours, 37000 Tours, France; 7Centre d’Etudes des Pathologies Respiratoires (CEPR), Institut National de la Recherche Médicale (INSERM), Unité Mixte de Recherche (UMR) 1100, Université de Tours, 37032 Tours, France

**Keywords:** systemic sclerosis, interstitial lung disease, lung ultrasound, cardiopulmonary exercise testing

## Abstract

**Objective:** High-resolution computed tomography (HRCT) may lack sensitivity for the early detection of interstitial lung disease associated with systemic sclerosis (SSc-ILD). Lung ultrasound is an emerging technique for the diagnosis of SSc-ILD. This cross-sectional study aimed to describe the prevalence of ultrasound interstitial syndrome in SSc patients with normal HRCT and pulmonary function tests (PFT). **Methods:** Thirty SSc patients with normal HRCT, FVC > 80% predicted and DLCO > 70% predicted were included. Echocardiography and PFT including impulse oscillometry and cardiopulmonary exercise testing were performed. Lung ultrasound was analyzed by two blinded operators. Patients were classified into two groups, according to the presence or absence of ultrasound interstitial syndrome, defined as the sum of B-lines in all thoracic areas ≥10 and/or pleural line thickness >3 mm on at least one thoracic area and/or a pleural line irregularity score >16%. **Results:** Ultrasound interstitial syndrome was present in 12 patients (40%). Inter-reader agreement for the diagnosis of ultrasound interstitial syndrome defined by the Kappa coefficient was 0.93 (95%CI 0.79–1.00). Patients with ultrasound interstitial syndrome were younger (37 years vs. 53 years, *p* = 0.009), more often had pitting scars (*n* = 7/12 vs. 3/18, *p* = 0.045) and had lower FVC (102 vs. 110% pred, *p* = 0.009), TLC (114 vs. 122% pred, *p* = 0.042) and low-frequency respiratory system reactance (Xrs5 Z-score 0.16 vs. 1.02, *p* = 0.018), while pulmonary gas exchange was similar. **Conclusions:** Ultrasound interstitial syndrome was detected in 12/30 SSc patients with normal HRCT and PFT. Patients with ultrasound interstitial syndrome had differences in lung function consistent with reduced respiratory compliance, suggesting minimal and/or early suspected SSc-ILD.

## 1. Introduction

Systemic sclerosis (SSc) is a rare autoimmune systemic disease characterized by microcirculation abnormalities, fibrosis of the skin and internal organs and autoimmune features [1]. SSc-related diffuse interstitial lung disease (SSc-ILD) affects more than half of all patients and is the leading cause of death in SSc [2].

Although many risk factors of the SSc-ILD development and its progression have been found, the identification of patients at risk of developing ILD and progression still remains a challenge [3]. High-resolution computed tomography (HRCT) of the chest is the gold standard for diagnosing and quantifying SSc-ILD [4]. This technique based on radiation is not optimal for iterative use. There is a need to develop new non-irradiating techniques for the early diagnosis of SSc-ILD [5].

Lung ultrasound (LUS) is a valuable technique for ILD imaging. LUS is a current research topic in the SSc and is identified as a potential additional screening tool for the detection of SSc-ILD [6]. Ultrasound interstitial syndrome comprises the three main following signs: B-lines, irregularity of the pleural line and thickening of the pleural line. Each of these signs correlates with the presence and severity of SSc-ILD on HRCT and the severity of respiratory function impairment [7,8]. Recently, the potential of lung ultrasound for the early diagnosis of SSc-ILD was suggested by the observation that ultrasound interstitial syndrome was observed in two SSc patients with normal HRCT [9].

We hypothesized that lung ultrasound could enable the identification of patients with lung interstitial abnormalities, which could be predictive of SSc-ILD, under the threshold of detection by HRCT and pulmonary function tests (PFTs). The main objective of this study was to describe the prevalence of ultrasound interstitial syndrome in SSc patients with normal HRCT, spirometry and carbon monoxide transfer. The secondary objectives were to determine if ultrasound interstitial syndrome was associated with clinical and physiological impairment consistent with SSc-ILD at rest and exercise.

## 2. Methods

### 2.1. Patients and Inclusion

The whole cohort of 170 SSc patients followed at the rare diseases center of Tours University Hospital, France, was screened, and suitable patients were invited to participate in a cross-sectional, monocentric study. All SSc diagnoses, new-onset and older, were established by internists affiliated with the rare disease center and specializing in SSc. Thirty patients with a diagnosis of SSc without lung damage were included between 1 March 2021 and 31 June 2022. 

Patients were eligible in this cross-sectional study if (1) they had a diagnosis of SSc and also fulfilled the 2013 ACR/EULAR classification criteria [10], (2) were older than 18 years of age, (3) had absence of ILD on last HRCT, (4) had forced vital capacity (FVC) ≥ 80% of the predicted value (pred) and lung carbon monoxide transfer (DLCO) ≥ 70% pred and forced expiratory volume in 1 s (FEV1)/FVC ≥ 0.7 on last PFT, and (5) they were able to perform cardiopulmonary exercise testing (CPET).

For each patient, all tests (HRCT, LUS, PFT, CPET and echocardiography) were performed on the same day. Demographic data, antibody status (anti-centromere, anti-topoisomerase I antibodies) and patient history were obtained retrospectively from medical records. Other data were obtained prospectively. Skin fibrosis was graded with the modified Rodnan skin score (mRSS) on the day of inclusion.

Patients were finally included if, on the day of inclusion, they (1) had absence of ILD on HRCT, and (2) had FVC ≥ 80% pred and DLCO ≥ 70% pred and FEV1/FVC ≥ 0.7. 

Patients were not included if they had any other connective tissue disease, restrictive and/or obstructive respiratory diseases, had echocardiographic signs of pulmonary hypertension (PH) [11] or left heart disease, had a total smoking burden >10 pack years, any acute respiratory disease requiring hospitalization in the year prior to inclusion or lower respiratory tract infection within 90 days or a history of chronic disease potentially affecting CPET.

All patients gave written consent. Ethical board approval was obtained on the 28 January 2021 (CPP-SUD EST III n°2021-008 B). The study was registered with Clinicaltrials.org (NCT04725786).

### 2.2. Intervention

#### 2.2.1. Lung Ultrasound

Lung ultrasound was performed using a Sparq ultrasound system (Philips Healthcare, in operation since 26 October 2018) with a convex probe (1–5 MHz) and one certified operator, blind to all other exams results. Lung ultrasound was practised according to a predefined protocol exploring 14 thoracic areas [12] (Appendix A) in right and left lateral decubitus. Each thoracic area was recorded for 6 s. All lung ultrasound images were reviewed by a blinded independent certified reader to measure inter-reader variability. A third reading was organised in the event of a ultrasonographer disagreement regarding ultrasound interstitial syndrome diagnosis. For each thoracic area, B-lines were counted, the pleural line was measured at the thickest location and the irregularity of the pleural line was determined as yes or no (Appendix A). B-lines were defined as vertical hyperechoic reverberation artefacts from the pleural line, extending to the bottom of the screen without attenuation and moving with breathing. Pleural line corresponds to ultrasound reflection at the level of the area grouping parietal pleura, visceral pleura and adjoining lung. Irregularity of pleural line was defined as loss of the linear contour of hyperechoic pleural line and was graded by pleural line irregularity score, as described by Pinal-Fenandez et al. [8]. 

The ultrasound interstitial syndrome was defined as the presence of at least one on the three positive ultrasound criteria and the sum of B-lines in all thoracic areas ≥10 [13] and/or thickness of the pleural line >3 mm on at least one thoracic area [7], and/or the pleural line irregularity score >16% [8]. The patient was considered to have ultrasound interstitial syndrome if they presented at least one of the three signs. These three criteria have been used in other interstitial lung pathologies [14].

To facilitate description of the distribution of ultrasound abnormalities, thoracic areas were grouped. The anterosuperior thoracic group comprised the para-sternal, mid-clavicular and anterior axillary areas. The posterobasal thoracic group comprised the mid-axillary, posterior-axillary, sub-scapular and paravertebral areas.

#### 2.2.2. Echocardiography and High-Resolution Computed Tomography

A cardiologist performed echocardiography. All the echocardiographies were conducted with Philips ultrasound system IE33. The TM and two-dimensional acquisition modes were used for the measurements of the cavities and axes. The flows were measured using Doppler mode. The evaluated parameters were dimension of cardiac cavities, systolic function and filling of each cardiac ventricle. The left ventricle ejection fraction (LVEF) was measured using the Simpson method.

Patients were excluded if echocardiography revealed left heart disease or maximum tricuspid flow velocity > 2.8 m·s^−1^ or indirect signs of pulmonary hypertension (right ventricle/left ventricle basal diameter ratio > 1.0, left ventricular eccentricity index > 1.1 in systole and/or diastole, right ventricular outflow Doppler acceleration time < 105 ms and/or midsystolic notching, early diastolic pulmonary regurgitation velocity >2.2 m/s, inferior cava diameter > 21 mm with decreased inspiratory collapse < 50% with a sniff or <20% with quiet inspiration, right atrial area > 18 cm^2^, PA diameter > 25 mm) [11].

Non-contrast-enhanced thin-slice chest HRCTs were acquired in the supine position at sustained end inspiration and a supplementary prone acquisition at end inspiration to clear position-induced changes. These examinations were performed either on a Siemens SOMATOM Definition Edge (Siemens Healthineers, Erlangen, Germany) with a tube voltage of 120 kV, adaptive tube current modulation, collimation of 2 × 64 × 0.6 mm with a z-flying focal spot, rotation time of 0.6 s, pitch 1.2, or a Toshiba Aquilion PRIME (Toshiba Medical Systems Europe, Amstelveen, The Netherlands) with a tube voltage of 120 kV, adaptive tube current modulation, collimation of 0.5 × 80 mm, rotation time of 0.4 s, pitch 1.38. Images were reconstructed using iterative reconstruction algorithms on both scanners, with a pulmonary kernel, a 512 × 512 matrix, a field of view adapted to patient morphology, with 0.6 mm thin slices on the Siemens scanner and 0.5 mm thin slices on the Toshiba scanner. 

Each HRCT was analysed by 2 researchers. Patients were excluded if HRCT revealed abnormalities, such as honeycombing, reticulations, ground-glass opacities, non-emphysematous cysts or traction bronchiectasis.

#### 2.2.3. Pulmonary Functional Test and Cardiopulmonary Exercise Testing

FVC and FEV1 were measured by spirometry. Total lung capacity (TLC) and functional residual capacity (FRC) were measured by plethysmography. DLCO was measured by the single-breath method and was corrected for hemoglobinemia (DLCOc). Impulse oscillometry (IOS) was used to measure respiratory system resistance at 5 and 20 Hz (Rrs5 and Rrs20), respiratory system reactance at 5 Hz (Xrs5), resonance frequency (Fres) and the area under the reactance curve (AX). All PFTs were conducted according to ATS and ERS guidelines [15]. Spirometry, plethysmography and DLCO measurements were expressed as percentages of the predicted value calculated using Global Lung Function Initiative predicted equations [16,17]. IOS measurements were expressed as Z-scores using Oostveen’s reference values [18].

Symptom-limited incremental CPET was performed blinded to lung ultrasound data according to ATS guidelines [19] using a cycle ergometer (Schiller, Germany) and the Ergocard metabolic cart (Medisoft, Belgium). Tests consisted of a 2 min rest stage, followed by an incremental work period at a slope of 10–20 W/min rate, and a 3 min recovery stage. Inspiratory capacity was measured every 2 min. FEV1 was measured before CPET and at the end of the recovery stage. Transcutaneous CO_2_ partial pressure (PtcCO_2_) and peripheral capillary oxygen saturation (SpO_2_) were monitored with a TCM5 sensor (Radiometer, Copenhagen, Denmark) positioned at the right earlobe [20]. PtcCO_2_-based Vd/Vt was calculated by the Bohr–Enghoff equation with correction for instrument dead space [21]. 

### 2.3. Number of Subjects Required

Based on the observation of fibrotic lung lesions in >40% of SSc patients in autopsy series [2,22], the expected prevalence of the suspected ultrasound interstitial syndrome was 30%. From this assumed prevalence, it was estimated that by recruiting 30 patients, we expected to observe an ultrasound interstitial syndrome in 9 patients.

### 2.4. Statistical Analysis

The primary aim of this study was to describe the prevalence of ultrasound interstitial syndrome in SSc patients with normal HRCT and lung function. Secondary objectives were to determine if patients with ultrasound interstitial syndrome had distinct clinical features and altered lung mechanics and pulmonary gas exchange in comparison with patients without ultrasound interstitial syndrome.

Continuous variables were expressed as median and interquartile range (1st quartile, 3rd quartile) and were compared with Mann-Whitney’s U-test. Categorical variables, expressed as total and proportion, were compared by Fisher’s exact test or Pearson’s Chi-squared test in the same grouping variable, as appropriate. Inter-reader agreement was assessed using kappa coefficients and intra-class correlation coefficient (ICC), as appropriate. Associations between the presence of ultrasound interstitial syndrome status and exercise data were assessed by a linear regression model adjusted for age.

Analyses were performed using R software, version 4.1.1 (R Foundation for Statistical Computing, Vienna, Austria). All statistical tests were fixed at a significance threshold of 5%. No intermediate analysis was conducted.

## 3. Results

### 3.1. Baseline Characteristics

Among the 170 SSc patients screened at our centre, 32 fulfilled the eligibility criteria. One patient refused to participate; two patients were not included because they had DLCO < 70% pred on the day of the study visit. Consequently, 30 patients were analysed (Appendix A).

The study population included 27 women (90%). Patient characteristics are reported in Table 1. Median (IQR) age at SSc onset was 48 (38, 58) years. Median follow-up since diagnosis was 8 (1, 12) years. Patients had almost exclusively limited cutaneous SSc (*n* = 29, 97%). Anti-centromere antibodies were detected in 24 patients (80%). The median modified Rodnan skin score (mRSS) was 5 (3, 6). All patients were Caucasian. 

### 3.2. Diagnosis of Ultrasound Interstitial Syndrome and Inter-Reader Agreement

Twelve patients (40%) had an ultrasound interstitial syndrome according to the first reader and eleven patients (37%) based on the second reader. After proofreading of the two operators, 12 patients (40%) had ultrasound interstitial syndrome. Details of lung ultrasound findings are reported in Table 2. Only two patients presented two signs of ultrasound interstitial syndrome, and none of them had all three signs. The median (95%CI) duration of lung ultrasound was 5 (95%CI: 5, 6) min. 

A description of the location of ultrasound abnormalities in patients with ultrasound interstitial syndrome is available in Appendix A and Figure 1. In patients with ultrasound interstitial syndrome, ultrasound abnormalities were observed predominately in postero-basal thoracic areas.

The Kappa coefficient illustrating inter-reader agreement for ultrasound interstitial syndrome was 0.93 (0.79 to 1.00). Kappa coefficients and interclass correlation coefficients for each individual lung ultrasound sign are reported in Table 3.

### 3.3. Association between Ultrasound Interstitial Syndrome and Lung Function 

Patients with ultrasound interstitial syndrome were younger at the time of the study and at the time of SSc diagnosis, had a shorter time from onset of Raynaud’s phenomenon to SSc diagnosis and had more frequent pitting scars (*n* = 7/12 vs. 3/18, *p* = 0.045)

Significant differences in lung function were observed between patients with or without ultrasound interstitial syndrome. Patients with ultrasound interstitial syndrome had a lower FVC (102 vs. 110% pred, *p* = 0.009), TLC (114 vs. 122% pred, *p* = 0.042) and FEV1 (99 vs. 108% pred, *p* = 0.010). They also had lower raw Xrs5 (−0.13 vs. −0.10 kPa·s·L^−1^, *p* = 0.046), lower Z-scores for Xrs5 (0.16 vs. 1.02, *p* = 0.018) and higher Z-scores for the area of respiratory system reactance AX (0.200 vs. −0.87, *p* = 0.016) and resonance frequencies (Fres) (3.11 vs. 2.57, *p* = 0.049). There was no difference in DLCOc.

At CPET, patients with ultrasound interstitial syndrome did not have reduced exercise capacity or alteration in pulmonary gas exchange compared to patients without ultrasound interstitial syndrome. No patients had dynamic hyperinflation or exercise-induced bronchoconstriction (Figure 2).

Because patients with ultrasound interstitial syndrome were markedly younger than patients without ultrasound interstitial syndrome, additional analyses were conducted to adjust the comparison of CPET data for age. After adjustment for age (Table 4), the peak workload ventilatory reserve was negatively associated with ultrasound interstitial syndrome status. By contrast, there was no significant association between ultrasound interstitial syndrome status and either peak load, peak VO_2_, the VE/VCO_2_ slope, PtcCO_2_-based Vd/Vt at VT1 or SPO_2_ at peak exercise. 

## 4. Discussion

The key finding of this pilot study was that lung ultrasound interstitial syndrome was present in 12 out of 30 SSc patients with normal HRCT, spirometry and DLCO. 

Patients with ultrasound interstitial syndrome had lower lung volumes (FVC, FEV1, TLC), low-frequency respiratory system reactance (Xrs5, AX, Fres) and a lower age-adjusted ventilatory reserve at exercise in comparison with patients without ultrasound interstitial syndrome. There are minor differences between patients with ultrasound interstitial syndrome and without that suggest a possible, via infra-clinical and infra-tomography, functional impairment, detectable by the CPET. These results are consistent with reduced respiratory system compliance [23,24] and support the hypothesis that ultrasound interstitial syndrome could be related to early SSc-ILD in this patient population. Although invasive measurements would be required to prove that lung ultrasound abnormalities were associated with reduced lung compliance, the lack of extensive skin sclerosis and chest sclerosis in all patients suggests that chest wall compliance was probably normal in the present study population. While reduced lung volumes were observed in patients with ultrasound interstitial syndrome, no difference was observed in terms of pulmonary gas exchange as there was no difference in DLCOc at rest and either SpO_2_ or Vd/Vt at exercise. It is possible that the lack of a significant difference in DLCO was due to insufficient study power since the measurement variability in DLCO is higher than the lung volumes [25], and such slight differences in lung function, as evidenced between patients with or without ultrasound interstitial syndrome, may be undetectable by CPET. Clinical correlation over time is indeed necessary to confirm these results.

The 12 patients with ultrasound interstitial syndrome were younger, had disease of more recent onset and more frequent pitting scars. These findings are consistent with the typical early onset of ILD during the course of SSc [26] and with the higher prevalence of ILD in patients with pitting scars in the EUSTAR cohort (48% vs. 29%) [27]. It is surprising that ultrasound interstitial syndrome was observed in a population of patients with predominant anti-centromere antibodies. The higher proportion of anti-centromere antibodies can be explained by the design of the study with the inclusion and exclusion criteria. We excluded all patients who already had ILD proven by the gold-standard HRCT and, therefore, almost all usual risk factors. Although ILD is typically associated with positive anti-topoisomerase I antibody, male gender and occupational exposure [28,29,30], recent data show that ILD can occur in patients with limited cutaneous SSc, anti-centromere antibodies and late disease. Indeed, Nihtyanova et al. estimated that, among 654 patients with clinically significant ILD, 139 had anti-centromeres antibodies, and 328 had limited cutaneous SSc [26].

The field of lung ultrasound is currently expanding in respiratory medicine [31], especially in the connective tissue disease [32]. Lung ultrasound is not radiation-based and is easily accessible. In the present study, the median time for performing lung ultrasound was 5 min. Furthermore, high inter-reader agreement confirms that lung ultrasound can be readily implemented in the clinic for routine practice.

Previous studies have reported the association between ultrasound interstitial syndrome signs and the severity of HRCT-defined SSc-ILD [33]. Our result is consistent with a previous study [9] and raises the hypothesis that lung ultrasound may be more sensitive than HRCT for the diagnosis of SSc-ILD, although a histopathological gold standard would be required to definitely reach such a conclusion. Indeed, 40 to 100% of patients with SSc are found to have some degree of parenchymal involvement in autopsy series [22,34,35], whereas HRCT-defined SSc-ILD is reported in 19 to 52% of patients [36,37]. Pulmonary damage is predominantly postero-basal in SSc-ILD [38]. In our study, ultrasound abnormalities predominated in the posterior lung areas, strengthening the hypothesis that lung ultrasound is sensitive for the detection of SSc-ILD. 

A recent meta-analysis reported promising diagnostic performance of ultrasound interstitial signs for SSc-ILD [6]. B-lines are present in all interstitial syndromes (e.g., pulmonary edema, pneumonia or the acute respiratory distress syndrome), and this is why we excluded patients with such symptoms. In a “healthy” population, B-lines can be found. A recent study characterized these non-pathological B-lines; they are less than or equal to 2, unilateral or very asymmetrical [39]. The B-line cut-off considered pathological differs depending on studies, and several authors adopt a B-line threshold < 10 [40]; however, we pursued greater specificity, hence adopting a more stringent criterion, consistent with other authors [6]. In our study, B-lines were considered as pathological if the sum of B-lines in all thoracic areas was ≥10, since the detection of 10 B-lines is highly predictive for the presence of significant SSc-ILD according to Tardella et al. [13]. Furthermore, B-lines were always bilateral and symmetrical.

Abnormalities of the pleural line may be the most specific of SSc-ILD. Pleural thickness > 3 mm is associated with reticulations on HRCT in SSc-ILD patients, with a sensitivity of 80% and a specificity of 99% [7]. The ultrasound sign “pleural line” does not correspond to the pleura or the subpleural space. Its abnormalities reflect structural abnormalities of the lung parechyma. The pleural line irregularity score is strongly associated with the Warrick ILD HRCT score [8]. The recent study of Beigi et al. confirms correlation between pleural line score and disease extension on quantitative chest tomography [41]. In our study, the most frequently occurring ultrasound sign was pleural line irregularity (5 (16.7%) patient reader 1; 8 (26.7%) patients reader 2). To our knowledge, this study is the first to use the three criteria comprising the sum of B-lines, irregularity of pleural line and/or thickness of pleural line for the detection of signs of SSc-ILD. These three criteria have already been used for idiopathic pulmonary fibrosis [14].

The main limitations of this study were the small sample size, heterogeneity in disease duration, the cross-sectional design and the absence of a group control. These limitations preclude any conclusions on the clinical significance of our findings. In particular, it is unclear whether patients with ultrasound interstitial syndrome will develop clinically significant SSc-ILD in the future.

The strength of our study is the use of three main signs to maximize the sensitivity of lung ultrasound for the diagnosis of interstitial abnormalities and the assessment of respiratory capacity, both at rest and during exercise. The single-center design reduced the transferability of the results but allowed for optimal consistency in patient characterization.

It is necessary to continue the explorations, notably with a longitudinal study, to evaluate the evolution over time of identified patients witch suspected SSc-ILD and in comparison to a healthy control group.

## 5. Conclusions

More than a third of SSc patients with normal HRCT, spirometry and DLCO had ultrasound interstitial syndrome. These patients appeared to have functional abnormalities, suggesting a slight alteration in respiratory system mechanics. Overall, and with caution, these data raise the hypothesis that lung ultrasound may have high sensitivity for the early diagnosis of suspected SSc-ILD. A longitudinal study evaluating the time course of patients with ultrasound interstitial syndrome and suspected SSc-ILD is needed to establish whether patients will develop symptomatic SSc-ILD in the future.

## Figures and Tables

**Figure 1 jcm-13-02885-f001:**
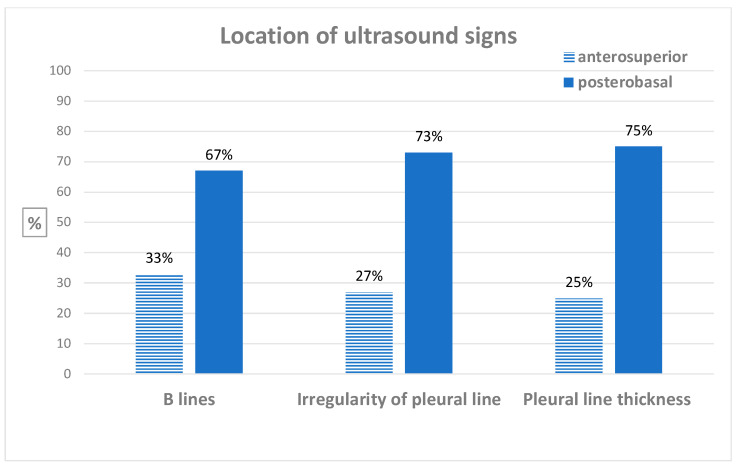
Location of ultrasound signs.

**Figure 2 jcm-13-02885-f002:**
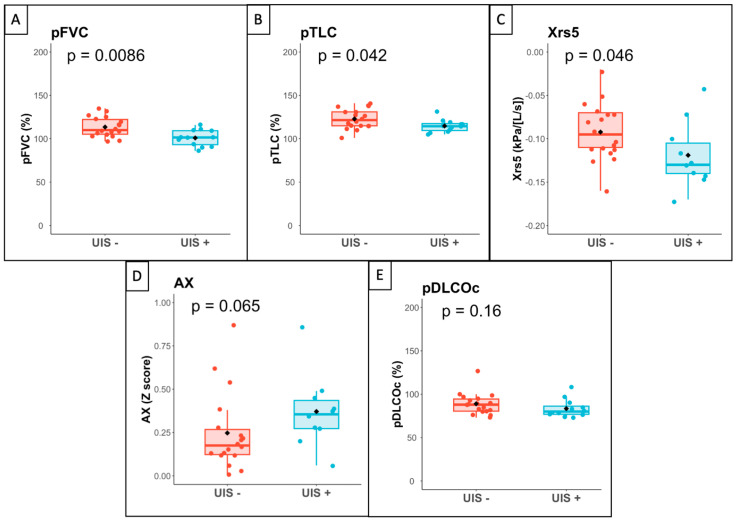
Boxplot for lung function data. Legend: Boxplot of lung function variables according to the presence (UIS+) or absence (UIS−) of ultrasound interstitial syndrome. (**A**) FVC: forced vital capacity (% predicted); (**B**) TLC: predicted total lung capacity (% predicted); (**C**) Xrs5: respiratory system reactance at 5 Hz (Z-score); (**D**) AX: Area of reactance (Z-score); (**E**) DLCO: hemoglobin-corrected lung transfer coefficient for carbon monoxide (% predicted). *p*-values calculated with Wilcoxon’s rank sum test.

**Table 1 jcm-13-02885-t001:** Patient characteristics.

	Overall N = 30	Patients with Ultrasound Interstitial Syndrome N = 12	Patients without Ultrasound Interstitial Syndrome N = 18	*p*
Sex: Female	27 (90%)	11 (92%)	16 (89%)	0.999
Age (years)	58 (46, 64)	47 (36, 54)	60 (58, 67)	0.003
Age onset (years)	48 (38, 58)	37 (22, 50)	53 (43, 59)	0.009
Time from Raynaud phenomenon to SSc (months)	26 (4, 148)	9 (2, 26)	114 (14, 294)	0.016
mRSS	5 (3, 6)	5 (4, 7)	5 (3, 5)	0.617
Puffy fingers	21 (70%)	6 (50%)	15 (83%)	0.102
Sclerodactylia	9 (30%)	6 (50%)	3 (17%)	0.102
Digital ulcer	2 (7%)	2 (17%)	0 (0%)	0.152
Pitting scars	10 (33%)	7 (58%)	3 (17%)	0.045
Telangiectasias	23 (77%)	9 (75%)	14 (78%)	0.999
Anti-centromere Antibodies	24 (80%)	10 (83%)	14 (78%)	0.999
Anti-topoisomerase I Antibodies	3 (11%)	1 (10%)	2 (11%)	0.999
**Pulmonary function tests**				
FEV1 (% pred)	108 (101, 116)	99 (92, 106)	108 (104, 122)	0.010
FVC (% pred)	106 (99, 111)	102 (93, 109)	110 (105, 122)	0.009
TLC (% pred)	117 (113, 127)	114 (110, 118)	122 (115, 131)	0.042
DLCOc (% pred)	84 (78, 92)	80 (77, 86)	88 (80, 94)	0.162
Xrs5 (Z-score)	−0.11 (−0.13, −0.07)	0.16 (−0.41, 0.82)	1.02 (0.76, 1.34)	0.018
Rrs5 (Z-score)	0.84 (0.26, 1.22)	0.59 (0.41, 1.06)	0.21 (0.02, 0.81)	0.44
AX (Z-score)	−0.64 (−1.31, 0.22)	0.200 (−0.46, 0.43)	−0.87 (−1.54, −0.59)	0.016
Fres (Z-score)	2.72 (2.47, 3.24)	3.11 (2.78, 3.32)	2.57 (2.41, 2.94)	0.049

Results are either numbers and percentages or median and interquartile range. *p*-values calculated with Fisher’s exact test, Pearson’s Chi-squared test or Wilcoxon’s rank sum test. Abbreviations: mRSS: modified Rodnan skin score; FEV1: forced expiratory volume in 1 s; FVC: forced vital capacity; TLC: Total Lung Capacity; DLCO: hemoglobin-corrected lung transfer coefficient for carbon monoxide; Xrs5: respiratory system reactance at 5 Hz; Rrs5: respiratory system resistance at 5 Hz; AX: area under the reactance curve; Fres: resonance frequency of the respiratory system.

**Table 2 jcm-13-02885-t002:** Lung ultrasound characteristics.

	N = 30
**Reader 1**	
Number of B-lines	2.5 (1.0–6.0)
Number of patients with sum of B-lines > 10	4 (13.3%)
Pleural line irregularity score (%)	14.3 (7.1–13.3)
Number of patients with irregularity > 16%	5 (16.7%)
Pleural line thickness (mm)	1.5 (1.4–1.6)
Number of patients with thickness > 3 mm	3 (10.0%)
**Number of patients with ultrasound interstitial syndrome**	12 (40.0%)
**Reader 2**	
Number of B-lines	2.0 (1.0–4.0)
Number of patients with sum of B-lines > 10	3 (10.0%)
Pleural line irregularity score (%)	10.7 (7.1–19.6)
Number of patients with irregularity > 16%	8 (26.7%)
Pleural line thickness (mm)	1.6 (1.4–1.6)
Number of patients with thickness > 3 mm	2 (6.7%)
**Number of patients with ultrasound interstitial syndrome**	11 (36.7%)

**Table 3 jcm-13-02885-t003:** Inter-reader agreement for each lung ultrasound sign and categorical classification.

	Kappa Coefficient	ICC	95% CI
**Number of B-lines**			
Ultrasound interstitial syndrome based on B-lines	0.84	-	0.53 to 1.00
Absolute B line quotation	-	0.77	0.76 to 0.77
**Pleural irregularity**			
Ultrasound interstitial syndrome based on irregularity	0.71	-	0.41 to 1.00
Absolute irregularity quotation	-	0.57	0.56 to 0.59
**Pleural thickness**			
Ultrasound interstitial syndrome based on thickness	0.78	-	0.37 to 1.00
Absolute pleural thickness quotation	-	0.32	0.31 to 0.33
**Total ultrasound interstitial syndrome assessment**	0.93	-	0.79 to 1.00

Abbreviations: ICC: interclass correlation coefficient 95% CI: 95% confidence interval.

**Table 4 jcm-13-02885-t004:** Comparison of CPET data adjusted for age.

CPET Data	Coefficient	*p* ^2^
Peak load	16.8908	0.355
Peak VO_2_	0.2238	0.333
VE/VCO_2_ slope	−3.4572	0.166
Peak SpO_2_	0.1689	0.566
PtcCO_2_-based. Vd/Vt VT1	−0.0143	0.653
Ventilatory reserve	−16.7918	0.023
Vd/Vt VT1	−0.0285	0.329

^2^ Association between the UIS and CPET data assessed by a linear regression model adjusted on age. Abbreviations: CPET: cardiopulmonary exercise testing; peak SpO_2_: peripheral capillary oxygen saturation at peak exercise; PtcCO_2_: transcutaneous CO_2_ partial pressure; VO_2_: oxygen uptake; VCO_2_: carbon dioxide production; Vd/Vt at VT1: dead space ventilation at first ventilatory threshold; VE: ventilatory flow.

## Data Availability

The data presented in this study are available on request from the corresponding author.

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
