# Peer review of "High Prevalence of the Lung Ultrasound Interstitial Syndrome in Systemic Sclerosis Patients with Normal HRCT and Lung Function—A Pilot Study"

_jcm, 2024, doi:10.3390/jcm13102885_

Round 1

Reviewer 1 Report

Comments and Suggestions for Authors

I had the opportunity to review your manuscript titled "High prevalence of the lung ultrasound interstitial syndrome in systemic sclerosis patients with normal HRCT and lung function – A pilot study." While the paper presents a compelling perspective on the diagnostic methods for SSc-ILD, I remain skeptical about the correlation between the sonographic findings and actual pathological changes in the lungs due to SSc. Firstly, a significant portion of the participants were positive for anti-centromere antibodies, which are generally not linked to SSc-ILD. Secondly, the observed differences between the UIS- and UIS+ groups were minor and fell within the normal limits. Thirdly, the criteria for defining UIS, based on a composite criterion, diverges from methodologies used in prior research. Lastly, it is crucial to conduct longitudinal studies to determine if cases with UIS+ evolve into SSc-ILD over time. Regrettably, I must state that the manuscript does not currently achieve the necessary level of quality for publication.

Comments on the Quality of English Language

Quality of English language is acceptable.

Reviewer 2 Report

Comments and Suggestions for Authors

Dear Authors,

Thank you for your work and contribution to science. I would like to express some of my criticisms about your article below.

1-It is not appropriate to include the sentence specified in lines 90-91 in the method section. It would be more appropriate to include it in introduction or discussion.

2-I believe that the most important limitation of the study is the lack of a healthy or patient control group. In this study, lung involvement was detected by USG in 40% of patients with normal PFT values and no ILD on HRCT. It should have been determined whether this was different at least with a control group.

3-The discussion includes the sentence "It is remarkable that the ultrasound interstitial syndrome was observed in a population of patients with predominant anti-centromere antibodies".  The proportion of anti-centromere antibody positive patients in your patient population is quite high, but anti-centromere positivity was similar between the two groups in the study. In addition, patients with ultrasound interstitial syndrome are younger and the time from Raynaud's onset to disease development is shorter. Early SSC-ILD formation is mostly associated with dcSSc and anti-scl70. What do you attribute your different result to? You should consider this issue more thoroughly in the discussion.

4-I recommend a short paragraph in the discussion about the limitations and strengths of the study.

Best regards.

Reviewer 3 Report

Comments and Suggestions for Authors

1.    Line 52: “It is radiation-based and thus unsuitable for iterative use”. CT is often used and highly necessary for “iterative use” in SS despite it being “radiation-based”, for example in judging effectiveness of antifibrotics. So, by all means, it is not “unsuitable”. You mean to say that it is not optimal or ideal for “iterative use” because it is “radiation-based”.
2.    Line 72: “Patients were pre-included in this cross-sectional study” – you need to briefly document randomness of inclusion. Did you screen all SSc patients coming to “the rare diseases center of Tours University Hospital” or only a part of them? Which part?
3.    Line 72: “Patients were pre-included in this cross-sectional study” – when did this happen? Please offer a short indication of time.
4.    Line 72: “Patients were pre-included in this cross-sectional study” – the literature does not systematically use this term “pre-inclusion” which is rather intuitive than scientific. I suggest using a different wording (for example “eligible”).
5.    Line 72-73: “diagnosis of SSc according to the 2013 ACR criteria” – who diagnosed these patients with SSc? A single doctor? Each/many doctor/s? Rheumatologist/s? Please briefly state this.
6.    Line 72-73: “diagnosis of SSc according to the 2013 ACR criteria” – I brotherly remind the authors that there are NO DIAGNOSIS CRITERIA for SSc. SSc is diagnosed by every physician based on clinical, biological and imaging data. The 2013 are CLASSIFICATION CRITERIA, used for clinical trials in order to include a homogenous and typical sample of SSc patients. A patient can be diagnosed with SSc DESPITE the fact that the classification criteria are not met. Therefore, I would rephrase to say that those patients were diagnosed with SSc and that their diagnoses ALSO FULFILLED the classification criteria.
7.    Line 72-73: “diagnosis of SSc according to the 2013 ACR criteria” – please do not omit EULAR from these criteria, they are ACR and EULAR endorsed. Those guys probably lost a night or two to produce the criteria so it is fair to mention them.
8.    Methods – we appreciate the clarity of mind when reporting criteria by “1- , 2-, 3- etc.”, but please be reminded that this is an academic text, not a notebook. Please include them in an enumeration : ; a) or 1) if you must.
9.    Line 156: “30 were finally included” – you correctly report the number of patients you obtained in the Results section and them go on to report their imaging results. Call me old fashioned but my article-reading mind would first want to know who are you talking about, what are the patient’s characteristics? So, Table 3 and lines 176-179 are better placed at the start of the Results section.
10.    Line 177: “Anti-centromere” + Table 3 anti-topoisomerase – the Results mention these antibodies but the Methods section did not say anything about blood tests. Please update the Methos section.
11.    In the Methos section and at line 176 you state that you report continuous variables as “median (IQR)”, but Table 3 and lines 176-179 report age as “48 (38, 58)”, “Median follow up since diagnosis was 8 (1, 12)” and “(mRSS) was 5 (3, 6)” etc. Maybe you changed your mind during draft writing, but these are medians with minimum and maximum, not IQR (which a single number since it results from the subtraction of 2 quartiles, for example: age of 48 (7) years”). You can use either one to report non-normally distributed continuous variables, but be consistent.
12.    Table 3 reports p values for associations of dichotomous nominal variables (for example sex and UIS/non-UIS), therefore you must have tested it with chi square tests but these tests are not mentioned in the Methods section. Please update the statistics section from the Methods.
13.    Results and Table 3 do not mention anything on race. Were all patients Caucasian? SSc has race differences.
14.    You may consider adding subdivisions/subheadings in the Methods (for example, patients and inclusion, hrCT, lung ultrasound, statistics etc.) and Results (idem) section for better reading.
15.    Discussion: please add a small paragraph at the end in which you confess the limitations or weaknesses of the study.

Round 2

Reviewer 1 Report

Comments and Suggestions for Authors

As I wrote in the previous report, I suggest rejection of the manuscript. The authors have not appropriately responded to my concerns.

Comments on the Quality of English Language

None.

Reviewer 2 Report

Comments and Suggestions for Authors

Dear Authors,

Thank you for your revisions. I have no additional suggestions.